# MUS81 nuclease activity is essential for replication stress tolerance and chromosome segregation in BRCA2-deficient cells

Xianning Lai[1], Ronan Broderick[2], Valérie Bergoglio[3], Jutta Zimmer[1], Sophie Badie[1], Wojciech Niedzwiedz[2], Jean-Sébastien Hoffmann[3] & Madalena Tarsounas[1]

Failure to restart replication forks stalled at genomic regions that are difficult to replicate or contain endogenous DNA lesions is a hallmark of BRCA2 deficiency. The nucleolytic activity of MUS81 endonuclease is required for replication fork restart under replication stress elicited by exogenous treatments. Here we investigate whether MUS81 could similarly facilitate DNA replication in the context of BRCA2 abrogation. Our results demonstrate that replication fork progression in BRCA2-deficient cells requires MUS81. Failure to complete genome replication and defective checkpoint surveillance enables BRCA2-deficient cells to progress through mitosis with under-replicated DNA, which elicits severe chromosome interlinking in anaphase. MUS81 nucleolytic activity is required to activate compensatory DNA synthesis during mitosis and to resolve mitotic interlinks, thus facilitating chromosome segregation. We propose that MUS81 provides a mechanism of replication stress tolerance, which sustains survival of BRCA2-deficient cells and can be exploited therapeutically through development of specific inhibitors of MUS81 nuclease activity.

[1] Genome Stability and Tumourigenesis Group, Department of Oncology, The CR-UK/MRC Oxford Institute for Radiation Oncology, University of Oxford, Old Road Campus Research Building, Oxford OX3 7DQ, UK. [2] Division of Cancer Biology, Institute of Cancer Research, 123 Old Brompton Road, London SW7 3RP, UK. [3] Cancer Research Center of Toulouse, Université de Toulouse, Inserm, CNRS, UPS, Equipe labellisée Ligue Contre le Cancer, Laboratoire d'excellence Toulouse Cancer, 2 Avenue Hubert Curien, Toulouse 31037, France. Correspondence and requests for materials should be addressed to M.T. (email: madalena.tarsounas@oncology.ox.ac.uk).

Replication stress represents a major source of genome instability stemming from slow rates of DNA synthesis, aberrant origin firing and frequent stalling of replication forks[1]. Treatment with agents that interfere with DNA replication (for example, hydroxyurea, aphidicolin), as well as oncogene overexpression[2] are known to trigger replication stress. Eukaryotic cells are prone to low levels of replication stress during normal, unchallenged cell cycle conditions. For example, barriers to fork progression (for example, DNA inter-strand cross links, DNA/RNA hybrids, G-quadruplexes) obstruct replication and cause fork stalling. To circumvent this problem and complete genome duplication, cells have evolved mechanisms for stabilizing and/or restarting stalled forks, some of which are dependent on the tumour suppressor BRCA2.

BRCA2 is a multifaceted protein best known for its function in promoting assembly of RAD51 filaments during homologous recombination (HR) repair. BRCA2 is also required during DNA replication to protect stalled replication forks against nucleolytic degradation[3] and to repair collapsed replication forks through RAD51-dependent restart reactions[4]. In addition, BRCA2 plays a role during mitosis where it regulates the metaphase to anaphase transition by sustaining spindle assembly checkpoint (SAC) via BubR1 acetylation[5]. Upon BRCA2 gene deletion, primary cells succumb to spontaneous double strand break (DSB) accumulation and checkpoint activation, which channel cells into senescence and apoptosis[6,7]. Cancer cells lacking BRCA2, however, acquire additional mutations, for example in tumour suppressor genes such as p53, which together with upregulation of error-prone DSB repair pathways sustain replication and proliferation[8]. Therefore, tolerance of high levels of replication stress and endogenous DNA damage enable survival of BRCA2-deficient cancer cells.

In eukaryotic cells, replication fork progression requires MUS81, a structure-specific endonuclease that acts in complex with its evolutionarily conserved partners EME1 (in yeast and human cells) or MMS4 (in budding yeast)[9]. MUS81-dependent nucleolytic cleavage promotes HR-dependent restart of stalled forks[10–12]. In mouse and human cells, the negative effects of MUS81 inactivation on fork restart have been examined following replication stress induced with hydroxyurea[11,13,14]. In addition, MUS81 is required in aphidicolin-treated human cells for common fragile site (CFS) replication[14,15] and DNA synthesis during mitosis[16].

Replication fork stalling at genomic regions that are difficult to replicate or contain endogenous DNA lesions is a hallmark of BRCA2 deficiency. We therefore investigated the impact of MUS81 on DNA replication in cells lacking BRCA2. Our results demonstrate that loss of MUS81 triggers increased replication stress and reduced survival in BRCA2-deficient cells. These cells progress into mitosis with incompletely replicated DNA, visualized as multiple chromosome interlinks in anaphase. Moreover, BRCA2-deficient cells rely on MUS81 to continue DNA synthesis during mitosis, the absence of which causes severe chromosome segregation defects and G1 arrest. We propose that in cells lacking BRCA2, MUS81-dependent nucleolytic cleavage removes DNA bridges caused by under-replicated DNA and provides a mechanism to complete replication in mitosis.

## Results

**Replication defects in cells lacking MUS81 and BRCA2.** To determine whether MUS81 and BRCA2 cooperate during unchallenged DNA replication, we measured replication rates using DNA fibre assays in H1299 human cells upon inactivation of these factors (Fig. 1a and Supplementary Table 1). Consistent with previous reports, we found that MUS81 inhibition had no effect on fork progression[17], while BRCA2 abrogation using a doxycycline (DOX)-inducible shRNA significantly decreased replication speed[18]. Strikingly, MUS81 inhibition in BRCA2-deficient cells caused a further slowdown in replication fork progression. We further observed that replication fork slowdown could be rescued by ectopically expressing siRNA-resistant wild-type MUS81, but not a catalytically inactive version of MUS81 (D338A/D339A), indicating that MUS81 nuclease activity is required to sustain replication in BRCA2-deficient cells (Fig. 1a). Wild-type MUS81 was expressed ectopically at levels similar to the catalytically inactive MUS81 (Supplementary Fig. 1a). Similar reduction in fork velocity was observed using U2OS cells, in which MUS81 and BRCA2 were depleted using siRNAs (Supplementary Fig. 1b and Supplementary Table 2). To determine the potential contribution of replication fork stalling to this phenotype, we measured the ability of MUS81- and/or BRCA2-depleted cells to restart stalled forks (Supplementary Fig. 1c). Resumption of DNA synthesis following treatment with 2 mM hydroxyurea was significantly impaired in cells lacking MUS81 or BRCA2. Concomitant abrogation of BRCA2 and MUS81 led to similar levels of fork stalling as either mutant alone, suggesting that MUS81 and BRCA2 act in the same pathway of replication fork restart. Most likely, stalled replication forks cleaved by MUS81 become substrates for BRCA2-dependent restart reactions. We recognize, however, that cell exposure to CldU/IdU for 1 h may mask some replication fork restart events. Expression of wild-type MUS81, but not catalytically inactive MUS81 reversed the increased frequency of stalled replication forks in MUS81-depleted cells (Supplementary Fig. 1d). This indicates that MUS81 nuclease activity is necessary for the restart of stalled replication forks, consistent with previous reports[19].

Next we examined the impact of MUS81 inactivation on the proliferation and survival of BRCA2-deficient cells. Although MUS81 depletion did not affect proliferation rates of H1299 cells, loss of BRCA2 led to a defect in proliferation. Co-depletion of MUS81 and BRCA2 caused a striking reduction in cell proliferation, suggestive of a synthetic lethal interaction between the two factors (Fig. 1b). This proliferation defect could be reversed by expression of wild type MUS81, but not of the catalytically inactive version, indicating that MUS81 nuclease activity is necessary for proliferation in BRCA2-deficient cells (Supplementary Fig. 2a,b). A substantial reduction in proliferation of U2OS cells co-depleted of MUS81 and BRCA2 was also observed (Supplementary Fig. 3a,b). The negative impact on BRCA2-deficient cell proliferation was recapitulated by depletion of SLX4, a scaffold protein that recruits MUS81 and stimulates its nucleolytic activity[20–22] (Fig. 1c). Furthermore, clonogenic assays provided evidence for a synthetic lethal interaction between MUS81 and BRCA2. Concomitant loss of the two proteins in H1299, U2OS and Calu-6 human cells resulted in a significant decrease in cell survival relative to BRCA2 depletion alone (Fig. 1d and Supplementary Fig. 3c–e). A similar decrease in cell survival was observed in human Capan-1 cells[23], derived from a BRCA2-deficient pancreatic tumour, upon depletion of MUS81 (Fig. 1e and Supplementary Fig. 3f).

To determine whether the proliferation defect and reduced survival of MUS81, BRCA2-deficient cells was associated with an increase in cell death, we stained cells for the apoptosis marker Annexin V. We observed a small, but significant increase in the percentage of Annexin V-positive cells in the absence of MUS81 and BRCA2, compared to inactivation of BRCA2 alone (Supplementary Fig. 3g). The effect seems minor, suggesting that other forms of cell death apart from apoptosis are likely involved. Given that BRCA2 is a key player in HR repair, we next addressed whether the observed synthetic lethality can be extended to other components of the HR pathway. We thus measured proliferation

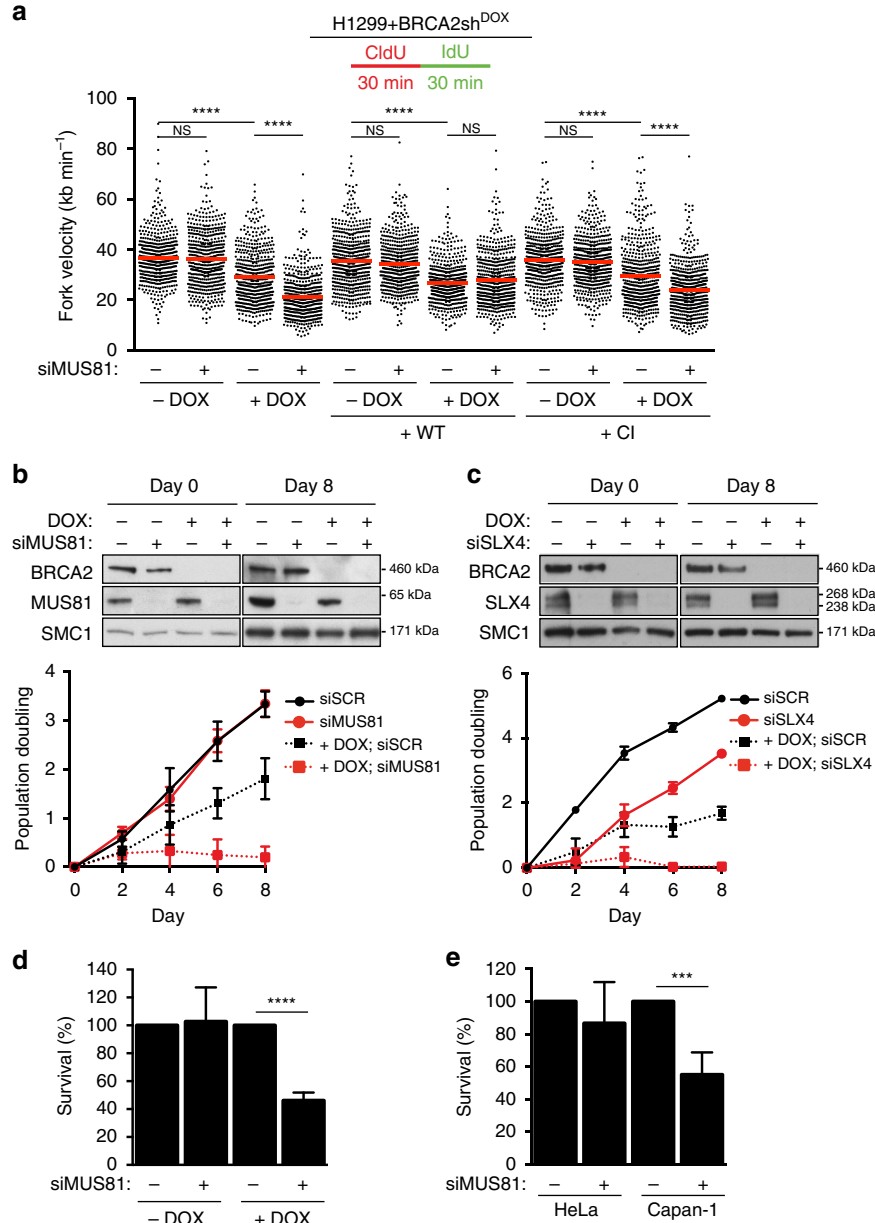

**Figure 1 | MUS81 facilitates replication and sustains proliferation in cells lacking BRCA2.** (**a**) H1299 cells carrying a doxycycline (DOX)-inducible BRCA2 shRNA were transfected with control or MUS81 siRNAs. Stable cells lines expressing either wild type (WT) or catalytically inactive (CI) human MUS81 were similarly processed. Cells were processed 24 h later for DNA fibre analysis as outlined in the inset, followed by quantification of CldU + IdU track length. Fork velocity was calculated using a conversion factor of $1 \mu m = 2.59 \, kb \, min^{-1}$. Red bars indicate mean ($n = 3$). ****$P < 0.0001$ (two-tailed Mann–Whitney test). (**b**) Cells treated as in **a** were processed 24 h later for proliferation assays. siRNAs were re-transfected at an interval of 4 days. Cell extracts prepared at indicated time points were immunoblotted as shown. SMC1 was used as a loading control. Graph shown is representative of three independent experiments. Error bars represent s.d. of triplicate values obtained from a single experiment. (**c**) H1299 cells expressing DOX-inducible BRCA2 shRNA were transfected with control or SLX4 siRNAs and processed as in **b**. The graph shown is representative of two independent experiments. Error bars represent s.d. of triplicate values obtained from a single experiment. (**d**) Cells treated as in **a** were plated 24 h later for clonogenic survival assays. Colonies were stained after 10–14 days. Error bars represent s.d. ($n = 4$). ****$P < 0.0001$ (unpaired two-tailed $t$-test). (**e**) HeLa or Capan-1 cells were transfected with control or MUS81 siRNAs and 48 h later plated for clonogenic survival assays. Colonies were stained after 10–14 days. Error bars represent s.d. ($n = 4$). ***$P < 0.001$ (unpaired two-tailed $t$-test).

rates of H1299 cells lacking BRCA1, RAD51 or RAD51C upon concomitant depletion of MUS81 (Supplementary Fig. 4a,c,e). Cells carrying MUS81 inactivation together with each of these HR factors displayed normal proliferation rates (Supplementary Fig. 4b,d,f). Collectively, these results suggest that the synthetic lethality between *MUS81* and *BRCA2* is unique among the HR genes. Most likely, BRCA2 specificity stems from its function in

sustaining SAC activation, which is not shared by BRCA1, RAD51 or RAD51C.

**Cells lacking BRCA2 activate DNA synthesis in mitosis.** Recent studies reported that cells that fail to complete replication during S phase continue replication in mitosis[24] and that this late DNA

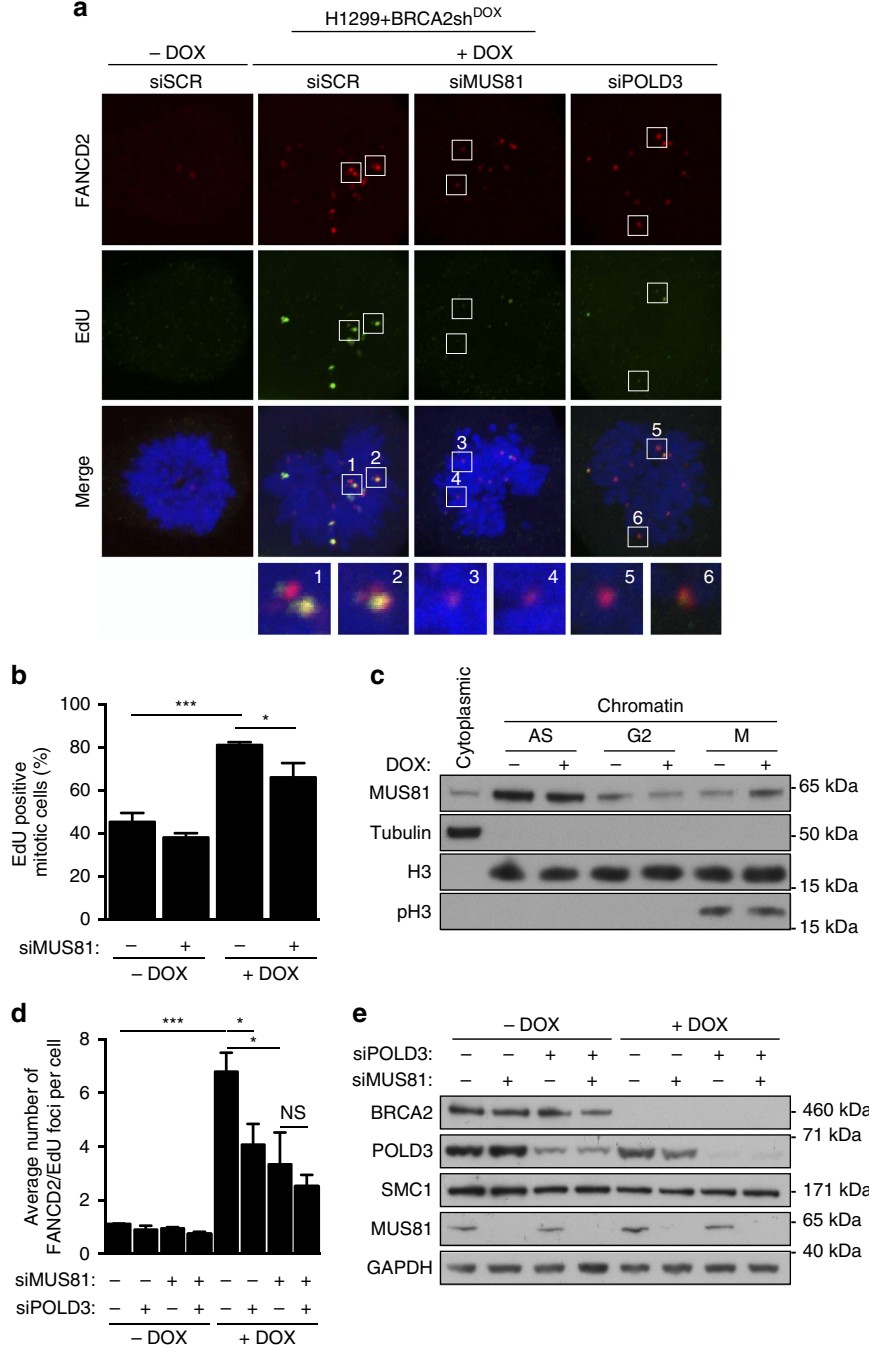

**Figure 2 | MUS81 promotes DNA synthesis during mitosis in BRCA2-deficient cells.** (**a**) H1299 cells carrying a DOX-inducible BRCA2 shRNA were transfected with control, MUS81 and/or POLD3 siRNAs. Cells were pulsed with EdU 72 h later and processed for detection of EdU (green) and immunofluorescence staining with anti-FANCD2 antibody (red). DNA was counterstained with DAPI. (**b**) Quantification of the frequency of EdU-positive mitotic cells treated as in **a**. Error bars represent s.d. ($n=3$). *$P<0.05$; ***$P<0.001$ (unpaired two-tailed $t$-test). (**c**) Western blot analysis of chromatin-bound fractions of indicated cell lines at indicated cell cycle stages. Histone H3 was used as a loading control for the chromatin fraction. AS, asynchronous; G2, G2-arrested; M, mitotic (prometaphase). (**d**) Quantification of the average number of FANCD2 foci co-localizing with EdU in mitotic cells treated as in **a**. Error bars represent s.d. ($n=3$). *$P<0.05$; ***$P<0.001$ (unpaired two-tailed $t$-test). (**e**) Cell extracts prepared from samples analysed in **d** were immunoblotted as indicated. SMC1 and GAPDH were used as loading controls.

synthesis is mediated by POLD3 (ref. 16). BRCA2-deficient cells exhibit slow replication fork progression and therefore are likely to enter mitosis with under-replicated DNA. To test whether these cells undergo mitotic DNA synthesis, we assessed EdU incorporation in mitotic chromosomes prepared from BRCA2-proficient and -deficient cells exposed to this nucleotide analogue (Fig. 2a), using conditions previously reported[24].

We observed a two-fold increase in the percentage of cells with EdU incorporation in BRCA2-deficient compared to BRCA2-proficient mitotic cells (Fig. 2b). Depletion of MUS81 in BRCA2-deficient cells led to a significant reduction in the percentage of EdU-positive cells (Fig. 2b). These results suggest that cells lacking BRCA2 continue DNA replication during mitosis and implicated MUS81 in this process. Consistent with

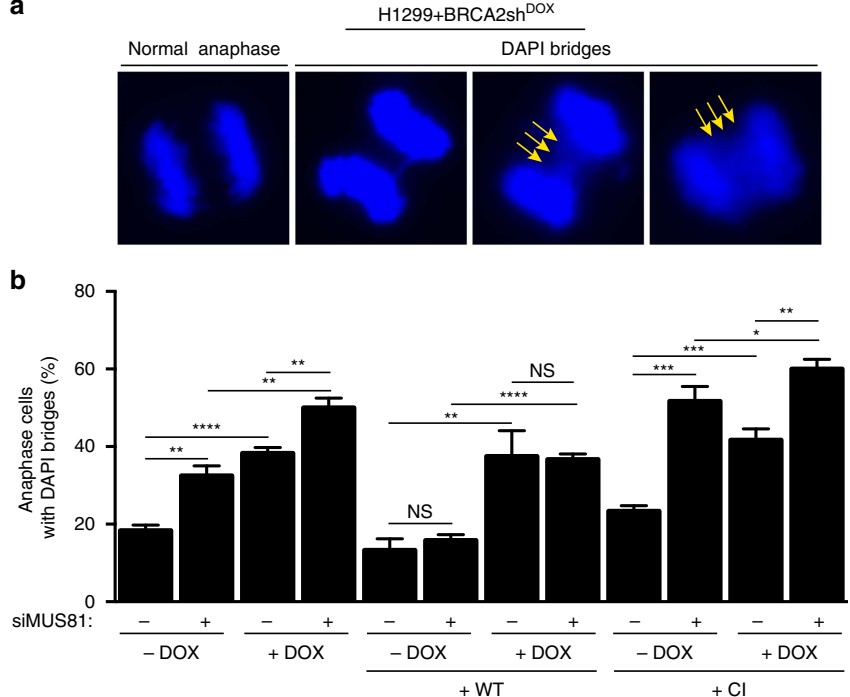

**Figure 3 | MUS81 resolves mitotic chromosome interlinks in BRCA2-deficient cells.** (**a**) H1299 cells carrying a DOX-inducible BRCA2 shRNA were transfected with control or MUS81 siRNAs and processed 72 h later for anaphase bridge analysis. Stable cells lines expressing either WT or CI human MUS81 were similarly processed. Representative images of DAPI-stained anaphase cells are shown. Yellow arrows indicate multiple DAPI bridges. (**b**) Quantification of the frequency of anaphase cells containing multiple DAPI bridges. Error bars represent s.d. ($n = 3$). *$P < 0.05$; **$P < 0.01$; ***$P < 0.001$; ****$P < 0.0001$ (unpaired two-tailed $t$-test).

this, we observed an increased recruitment of MUS81 to the chromatin of BRCA2-deficient mitotic cells (Fig. 2c), suggesting that BRCA2-deficient cells rely more on MUS81 for activating replication in mitosis than BRCA2-proficient control cells.

DNA synthesis at under-replicated CFS marked by FANCD2 was previously shown to require MUS81 and POLD3, a DNA polymerase involved in break-induced replication[16,25]. To assess the role of MUS81 and POLD3 during DNA synthesis in BRCA2-deficient mitotic cells, we quantified the number of EdU foci at CFS identified by FANCD2 co-localization. We observed an increase in EdU incorporation at CFS in mitotic cells lacking BRCA2 compared to BRCA2-proficient counterparts (Fig. 2d,e). This was dependent on MUS81 and POLD3 as the number of EdU foci co-localizing with FANCD2 was reduced upon depletion of either or both factors. Taken together, these results indicate that BRCA2-deficient cells perform DNA synthesis in mitosis and that MUS81 is required for this process. In addition, POLD3 is implicated in late DNA synthesis at CFS in these cells. However, concomitant depletion of MUS81 and POLD3 does not fully abolish mitotic DNA synthesis in BRCA2-deficient cells, indicating that a MUS81- or POLD3-independent pathway contributes to its activation. SLX4 is recruited to CFS in mitosis[16], where, together with its interacting nucleases, may promote an alternative mechanism for DNA synthesis initiation.

**Chromosome mis-segregation in cells lacking MUS81 and BRCA2.** Regions of under-replicated DNA associated with CFS pose problems during chromosome segregation and can be visualized as DAPI- or BLM-positive ultra fine bridges (UFBs) in anaphase cells[26,27]. To test whether MUS81 and BRCA2 co-depletion could impact on chromosome segregation, we quantified DAPI-positive anaphase bridges (Fig. 3a). H1299 human cells

lacking either BRCA2 or MUS81 exhibited an increase in the percentage of cells with multiple DAPI-stained bridges. Formation of anaphase bridges in BRCA2-deficient has been previously reported in mouse embryonic stem cells[28]. Strikingly, MUS81 inactivation in BRCA2-depleted cells led to a significantly higher percentage of cells with multiple DAPI-positive bridges relative to BRCA2 depletion alone (Fig. 3b). This increase was reversed by expression of wild-type MUS81, but not by expression of catalytically inactive MUS81, indicating that the nuclease activity of MUS81 was required to prevent accumulation of chromosome interlinks in BRCA2-deficient cells. Interestingly, expression of the catalytically inactive MUS81 induced a higher level of anaphase bridges than MUS81 siRNA-mediated depletion itself, consistent with a dominant negative effect of this variant. Binding of the catalytically inactive MUS81 to anaphase bridges may prevent access of other nucleases, which could resolve these structures in the absence of MUS81. Although we observed a slight increase in the frequency of DAPI-negative, BLM-positive UFBs in MUS81- or BRCA2-depleted cells, this was not further increased when both proteins were concomitantly inactivated (Supplementary Fig. 5a,b). The presence of multiple DAPI bridges in these cells could conceivably lower DNA tension and prevent UFB assembly[29].

**Cytokinesis failure in MUS81- and BRCA2-deficient cells.** Previous work[5] demonstrated that BRCA2 sustains the SAC by promoting BubR1 acetylation in human HeLa cells. Thus, premature SAC inactivation results in shorter mitosis in cells lacking BRCA2. We first sought to determine mitotic duration in H1299 cells in which we depleted BRCA2 using a DOX-inducible shRNA. We measured the time interval required for the progression from anaphase onset to cytokinesis using time-lapse microscopy (Supplementary Fig. 6a). In contrast to an average of

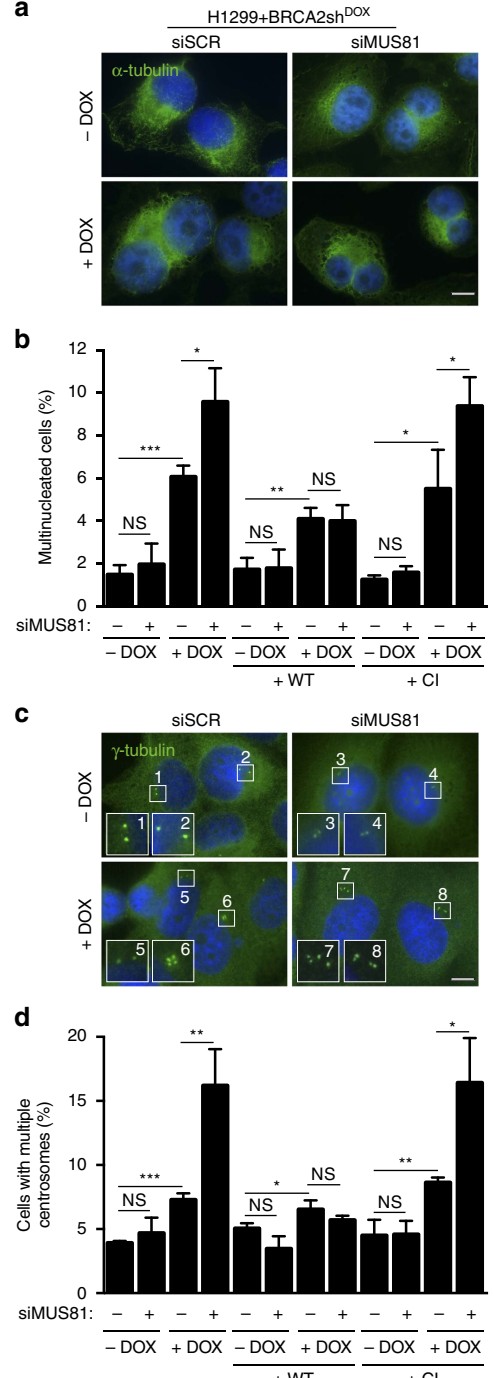

**Figure 4 | MUS81 inhibition leads to multinucleation and supernumerary centrosomes in cells lacking BRCA2.** (**a**) H1299 cells carrying a DOX-inducible BRCA2 shRNA were transfected with control or MUS81 siRNAs. Representative images of cells stained with an α-tubulin antibody (green) 72 h after transfection are shown. DNA was counterstained with DAPI. Scale bar, 10 μm. (**b**) Quantification of the frequency of multinucleated cells in asynchronous cultures treated as in **a**. Similar analyses were conducted using stable cells lines expressing either WT or CI human MUS81. Error bars represent s.d. ($n = 3$). *$P < 0.05$; **$P < 0.01$; ***$P < 0.001$ (unpaired two-tailed $t$-test). (**c**) Cells treated as in **a** were processed for immunostaining with γ-tubulin (green) 72 h after transfection. Insets show selected regions containing centrosomes at a higher magnification. Scale bar, 10 μm. (**d**) Quantification of the frequency of cells with multiple centrosomes treated as in **c**. Similar analyses were conducted using stable cells lines expressing either WT or CI human MUS81. Error bars represent s.d. ($n = 3$). *$P < 0.05$; **$P < 0.01$; ***$P < 0.001$ (unpaired two-tailed $t$-test).

2 h in BRCA2-proficient cells, BRCA2-deficient cells completed mitosis in approximately 1 h (Supplementary Fig. 6b,c). Consistent with this, the mitotic index measured by FACS analysis of phosphorylated histone H3 in asynchronous populations of BRCA2-deficient cells was significantly lower than in BRCA2-proficient controls (Supplementary Fig. 6d). MUS81 depletion did not affect either mitotic duration or the mitotic index. These results support a role for BRCA2 in regulating mitosis duration independent of MUS81.

The short mitosis duration in BRCA2-deficient cells, together with the elevated levels of DAPI-positive bridges suggested possible cytokinesis-related defects in these cells. Consistent with this notion, we observed a significant increase in the frequency of multinucleated H1299 cells in the absence of BRCA2, which was more pronounced upon concomitant depletion of MUS81 (Fig. 4a,b). The same results were obtained when similar assays were performed in U2OS cells (Supplementary Fig. 7a,b). Multinucleated cells arising from cytokinesis defects commonly display supernumerary centrosomes. Quantification of the percentage of cells with multiple centrosomes detected by γ-tubulin demonstrated a significant increase in the context of BRCA2 deficiency, which was further augmented by MUS81 co-depletion (Fig. 4c,d and Supplementary Fig. 7c,d). Both the high frequency of multinucleated cells and of cells with supernumerary centrosomes were reversed by expression of wild type MUS81, but not of the catalytically inactive MUS81 (Fig. 4b,d). This indicates that the nuclease activity of MUS81 is required to prevent cytokinesis failure in BRCA2-deficient cells.

**53BP1 bodies and G1 arrest in cells lacking MUS81 and BRCA2.** Replication stress elicits elevated levels of 53BP1 nuclear bodies in G1 cells[27,30], possibly marking DNA lesions inflicted during chromosome segregation in mitosis. To determine whether the mitotic abnormalities observed in BRCA2-deficient cells similarly resulted in DNA damage accumulation in G1, we measured the frequency of G1 cells containing 53BP1 nuclear bodies. To exclude non-G1 cells from our quantification, we counterstained cells with cyclin A, a marker for cells in S and G2 stages of the cell cycle. Notably, multinucleated cells were cyclin A-positive and therefore excluded from this analysis. We found a significant increase in the frequency of 53BP1 nuclear bodies in cells lacking MUS81 or BRCA2, which was further elevated when the two proteins were concomitantly depleted (Fig. 5a,b). The frequency of 53BP1 nuclear bodies was reversed to control levels by expression of wild-type MUS81, but not of catalytically inactive MUS81 (Fig. 5b). These results suggest that MUS81 nuclease activity is required to suppress G1 DNA damage accumulation in cells lacking BRCA2.

To establish whether DNA damage accumulation led to cell cycle defects, we carried out fluorescence-activated cell sorting (FACS) analysis of EdU-pulsed asynchronous cell populations. We observed that BRCA2 depletion caused an enrichment of cells in G1. Importantly, depletion of MUS81 in these cells lead to further increase in the frequency of G1 cells, accompanied by a reduction in the percentage of cells in S phase of the cell cycle (Fig. 5c). This supports the notion that elevated levels of chromosome mis-segregation and DNA lesions inflicted by aberrant mitosis elicit G1 arrest in BRCA2- and MUS81-deficient cells.

**Discussion**
DNA replication is regulated at multiple levels to ensure complete genome duplication during S phase and faithful chromosome segregation during mitosis. Mechanisms that regulate fork progression and protect its integrity prevent forks from stalling. Additional mechanisms ensure replication restart when fork

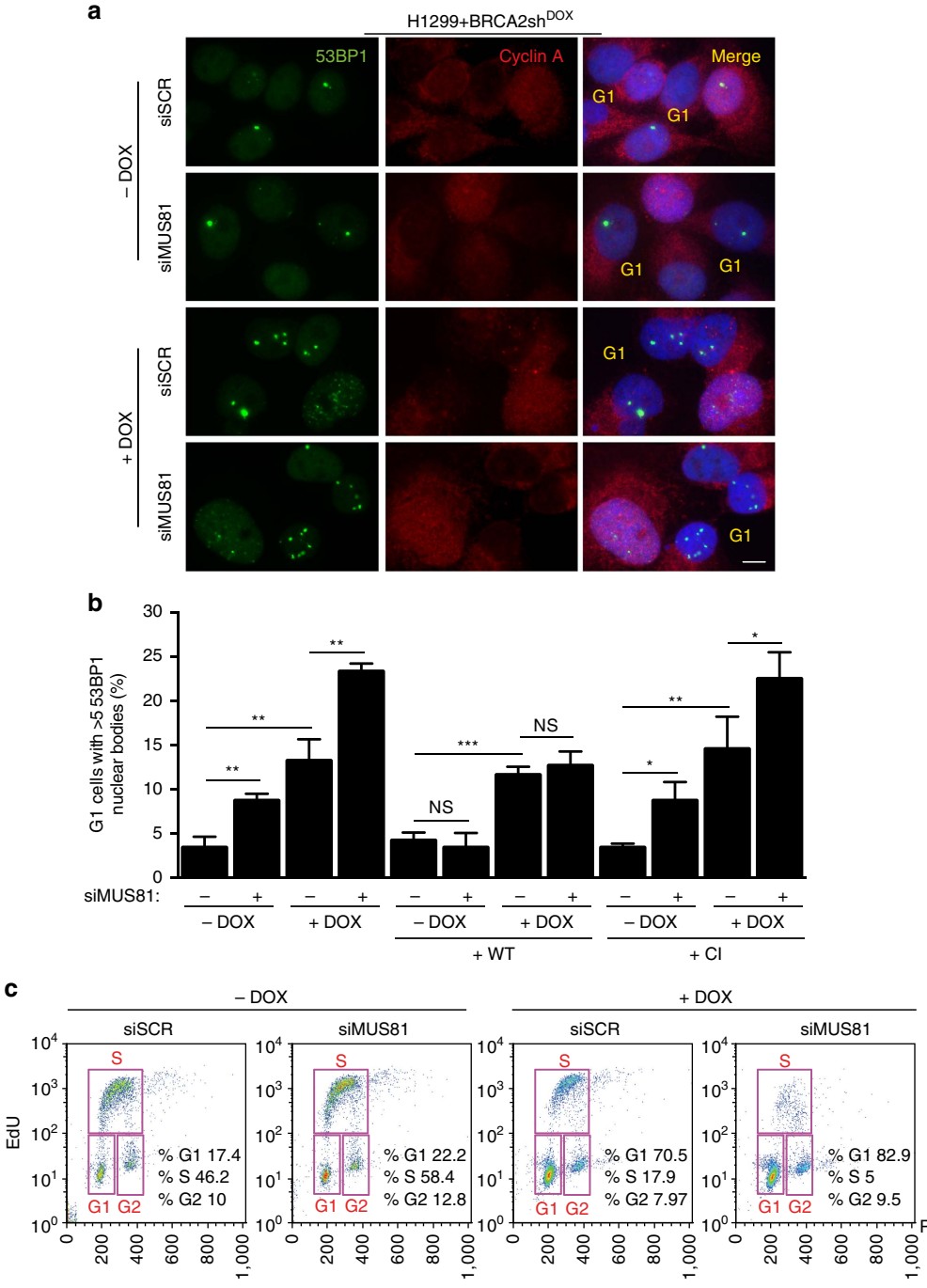

**Figure 5 | MUS81 inhibition in BRCA2-deficient cells causes accumulation of 53BP1 nuclear bodies and G1 arrest.** (**a**) H1299 cells carrying a DOX-inducible BRCA2 shRNA were transfected with control or MUS81 siRNAs. Representative images of cells processed 72 h later for immunostaining with anti-53BP1 (green) and anti-cyclin A (red) antibodies. DNA was counterstained with DAPI. Scale bar, 10 μm. (**b**) Quantification of the frequency of cyclin A-negative G1 cells containing > 5 53BP1 nuclear bodies in cells treated as in **a**. Similar analyses were conducted using stable cells lines expressing either WT or CI human MUS81. Error bars represent s.d. (*n* = 3). *$P < 0.05$; **$P < 0.01$; ***$P < 0.001$ (unpaired two-tailed *t*-test). (**c**) Quantification of G1, S and G2 cell populations (boxed) in asynchronous cultures of EdU-labelled cells treated as in **a**. PI, propidium iodide.

stalling occurs. BRCA2 plays essential roles both in the protection of the stalled replication forks by preventing their nucleolytic degradation[3] and in fork restart through RAD51-mediated reactions[31]. Recently, it has been reported that MUS81 facilitates DNA replication not only upon treatment with drugs that interfere with DNA replication[13,14,17], but also in the absence of exogenous damage[19]. Whether MUS81 and BRCA2 proteins act together or independently of each other during unchallenged DNA replication remained unclear.

Here we show that MUS81 sustains replication fork progression in BRCA2-deficient cells through mechanisms distinct from the restart of stalled replication forks. One possibility is that MUS81 cleaves and resolves DNA secondary structures ahead of the fork, preventing them from becoming barriers to advancing replication. Our results demonstrate that MUS81 abrogation does not increase the frequency of fork stalling in BRCA2-deficient cells, suggesting that MUS81 and BRCA2 act epistatically to restart replication. Conceivably, cleavage by MUS81 generates a

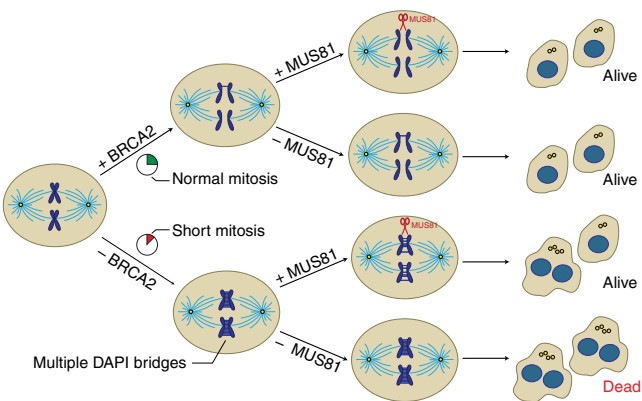

**Figure 6 | Model for concerted action of MUS81 and BRCA2 during mitosis.** BRCA2-proficient cells require MUS81 for cleavage of UFBs formed during mitosis at under-replicated CFS. In cells lacking BRCA2, incomplete DNA replication at multiple sites leads to DAPI-stained bridges detectable in anaphase as chromosome interlinks. MUS81 is required to resolve these bridges and to promote mitotic DNA synthesis, ultimately facilitating chromosome segregation. MUS81 inactivation in BRCA2-deficient cells leads to persistent chromosome interlinks, multinucleation, supernumerary centrosomes and cell death. Blue, sister chromatids; yellow, centrosomes; light blue, microtubules.

DNA structure that becomes a substrate for HR-dependent fork restart reactions.

Decreased rates of fork progression in cells lacking BRCA2 can lead to failure to complete genome replication prior to entry into mitosis. Recent work has shown that activating DNA synthesis during mitosis can compensate for unfinished replication[16,24]. Consistent with this, we detected increased EdU incorporation during mitosis in BRCA2-deficient cells. In addition to POLD3, MUS81 was required for late DNA synthesis in the absence of BRCA2. Consequently, depletion of MUS81 in BRCA2-deficient cells prevented activation of this compensatory mechanism, causing under-replicated DNA to persist until chromosome segregation in anaphase. Consistent with this notion, cells lacking MUS81 and BRCA2 showed strikingly high levels of DAPI-positive bridges at anaphase, indicating that MUS81 resolves structures that prevent anaphase bridge formation in BRCA2-deficient cells.

The severe catenation phenotype of the MUS81, BRCA2 co-depleted cells led to chromosome segregation defects reflected in increased multinucleation, supernumerary centrosomes and DNA lesions marked by 53BP1 bodies in G1. In spite of being p53-null and thus G1 checkpoint-impaired, these cells remained arrested in G1. It is conceivable that p53-independent mechanisms are activated to prevent transition from G1 to S phase[32,33]. Alternatively, factors that are required for S phase initiation may be deregulated as a consequence of chromosome mis-segregation.

Here we propose that MUS81 provides a mechanism of replication stress tolerance that sustains proliferation and survival of BRCA2-deficient cells. These cells exhibit constitutively high levels of endogenous replication stress, which is further exacerbated upon MUS81 inactivation. In cancer cells lacking BRCA2 (Fig. 6), dysfunctional checkpoints, including SAC failure, enable mitotic entry and progression with incompletely replicated genomes. This results in interlinks between sister chromatids in anaphase, which require the nucleolytic activity of MUS81 for their resolution, followed by late DNA synthesis, to enable correct chromosome segregation. Thus, BRCA2-deficient cells rely on MUS81 not only to reduce the replication stress burden during S phase, but also to eliminate the detrimental consequences of under-replicated DNA during mitosis.

Our discovery that MUS81 inactivation is lethal when combined with BRCA2 deficiency identifies MUS81 as a potential target for the development of drugs that could selectively eliminate BRCA2-compromised cells and tumours. In particular, the nuclease activity of MUS81, which we found to be required for attenuation of the replication and mitotic defects characteristic of BRCA2 deficiency, represents an attractive druggable target from a pharmacological point of view.

## Methods

**Cell lines and growth conditions.** Human non-small cell lung carcinoma H1299 cells, human osteosarcoma U2OS cells (both from American Type Culture Collection) and human cervical cancer HeLa OHIO cells (from Cancer Research UK Cell Services) were cultivated in monolayers in Dulbecco's Modified Eagle Medium (DMEM, Sigma) supplemented with 10% foetal bovine serum (Life Technologies), penicillin and streptomycin (Sigma). Human Calu-6 cells were cultivated in monolayers in Advanced DMEM/F12 (Thermo Fisher) supplemented with 15% foetal bovine serum (Life Technologies), L-Glutamine (Sigma), penicillin and streptomycin (Sigma). Human pancreatic cancer Capan-1 cells (from Cancer Research UK Cell Services) were cultivated in monolayers in Iscove's Modified Dulbecco's Medium (IMEM, Thermo Fisher) supplemented with 20% foetal bovine serum (Life Technologies), penicillin and streptomycin (Sigma). H1299 cells expressing doxycycline (DOX)-inducible BRCA1, BRCA2 or RAD51C shRNAs were established using the 'all-in-one' system[34]. shRNAs targeting BRCA1 5′-GAG TAT GCA AAC AGC TAT AAT CTC GAG ATT ATA GCT GTT TGC ATA CTC-3′, BRCA2 5′-GGG AAA CAC UCA GAU UAA A-3′ or RAD51C 5′-GAG AAU GUC UCA CAA AUA A-3′ were cloned into the pLKO$^{TetOn}$ plasmid and constructs were introduced into H1299 cells using lentiviral infection as described[35]. DOX-inducible H1299 cells were cultivated in monolayers in DMEM medium (Sigma) supplemented with 10% tetracycline free foetal bovine serum (Clontech), penicillin and streptomycin (Sigma). To isolate G2 cells, H1299 cells were treated with 9 μM of RO-3306 for 16 h. G2-arrested cells were subsequently released for 20 min in fresh medium to obtain mitotic populations.

**siRNAs.** H1299 and U2OS cells were transfected using Dharmafect 1 (Dharmacon). Briefly, $0.8 \times 10^6$ cells were transfected with 40 nM siRNA by reverse transfection in 10-cm plates. After 24 h incubation, depletion was evident as determined by immunoblotting. For long-term experiments such as proliferation assays, cells were re-transfected at an interval of 4 days. Sequences of siRNAs used were as follows: siSCR 5′-CGT ACG CGG AAT ACT TCG A-3′ (Dharmacon), siMUS81 5′-CAG CCC UGG UGG AUC GAU A-3′ (Dharmacon), siSLX4 5′-AAA CGT GAA TGA AGC AGA ATT-3′, siGENOME BRCA2 (Dharmacon), esiRAD51 (Sigma) and ON-TARGETplus POLD3 siRNA (Dharmacon).

**Lipofectamine transfection.** Wild type or catalytic inactive MUS81 expression construct in pcDNA3.3 vector (a gift from Dr Sheroy Minocherhomji, University of Copenhagen) was made siRNA-resistant by introducing silent point mutations within the siRNA target sequence. H1299 cells were transfected with expression plasmids of siRNA-resistant wild-type MUS81 or catalytic inactive MUS81-D338A-D339A using Lipofectamine 2000 (Invitrogen) according to manufacturer's instructions. Stable cell lines were established by selection in 0.5 mg ml$^{-1}$ G418.

**Cell proliferation assays.** Cells were plated at densities ranging between 1,000 and 2,000 cells per well in 96-well plates. Cell viability was determined by incubating cells with medium containing 10 μg ml$^{-1}$ of resazurin for 2 h. Fluorescence was measured at 590 nm using a plate reader (POLARstar, Omega one). To determine population doublings, resazurin-based readouts of cell viability were taken after cells had adhered (day 0) and at 48 h intervals for 8 days.

**Clonogenic assays.** Cells were plated at densities between 200 and 800 cells per well in 6-well plates. Colonies were stained with 5 mg ml$^{-1}$ crystal violet (Sigma) in 50% methanol and 20% ethanol. Cell survival was expressed relative to control cells.

**EdU incorporation in asynchronous cells.** To label replicated DNA, cells were incubated with 10 μM EdU for 45 min. Samples were collected by trypsinization and incorporated EdU was detected using the Click-iT EdU Alexa Fluor 647 Flow Cytometry Assay Kit (Molecular Probes) according to manufacturer's instructions. Cells were re-suspended in PBS containing 20 μg ml$^{-1}$ propidium iodide (Sigma) and 10 μg ml$^{-1}$ RNase A (Sigma) before samples were processed using flow cytometry (BD FACSCalibur, BD Biosciences). A number of 10,000 events were analysed per condition and experiment using FlowJo software.

**Annexin V assay.** Apoptotic and dead cells were labelled with a FITC Annexin V/Dead Cell Apoptosis Kit (Thermo Fisher Scientific) according to the manufacturer's instructions. Stained cells were immediately analysed by flow cytometry (FACSCalibur, BD Biosciences). Apoptotic cells bound Annexin V-FITC. A number of 20,000 events were analysed per condition and experiment with FlowJo software.

**Mitotic cell labelling.** Cells were incubated with an anti-phosphorylated histone H3 (1:200, Abcam) for 1 h at room temperature, followed by washes in PBS and incubation with fluorochrome-conjugated secondary antibody (goat anti-rabbit Alexa Fluor 488, 1:200, Molecular Probes) for 1 h at room temperature. Cells were washed and re-suspended in PBS containing 20 µg ml$^{-1}$ propidium iodide (Sigma) and 10 µg ml$^{-1}$ RNase A (Sigma), before processing by flow cytometry (BD FACSCalibur, BD Biosciences). A number of 10,000 events were analysed per condition and experiment using FlowJo software.

**Time-lapse microscopy and live cell imaging.** Time-lapse microscopy was performed using a Nikon TE2000-E motorized inverted microscope equipped with a complete incubation system (Nikon). Time-lapse images of H1299 cells were taken with the Orca-ER CCD digital camera (Hamamatsu Photonics) at 5-min intervals for 48 h. Anaphase onset was defined as the time the cell starts to round up. All images were processed using NIS Elements software (Nikon).

**Immunofluorescence.** Cells were washed in PBS, swollen in hypotonic solution (85.5 mM NaCl and 5 mM MgCl$_2$) for 5 min. Cells were then fixed with 4% paraformaldehyde for 10 min at room temperature and permeabilized by adding 0.03% SDS to the fixative. After blocking with blocking buffer (1% goat serum, 0.3% BSA, 0.005% Triton X-100 in PBS), cells were incubated with primary antibody diluted in blocking buffer overnight at room temperature. Then, they were washed again and incubated with fluorochrome-conjugated secondary antibodies (1:400, Molecular Probes) for 1 h at room temperature. For centrosome visualization, cells were washed in PBS and fixed with 100% pre-chilled methanol for 5 min at −20 °C. Coverslips were washed twice in PBS. After blocking (0.2% PBS-Tween, 0.1% BSA, 1% FBS) for 1 h at room temperature, cells were incubated with γ-tubulin antibody (1:2,000, GTU-88, Abcam) diluted in blocking buffer for 30 min at room temperature. Cells were washed with 0.1% Triton X-100 in PBS and incubated with fluorochrome-conjugated secondary antibodies (goat anti-mouse Alexa Fluor 488, 1:400, Molecular Probes) for 40 min at room temperature. Dried coverslips were mounted on microscope slides using the ProLong Antifade kit (Life Technologies) supplemented with 2 µg ml$^{-1}$ DAPI. Samples were viewed with a Leica DMI6000B inverted microscope and fluorescence imaging workstation equipped with a HCX PL APO × 100/1.4–0.7 oil objective. Images were acquired using a Leica DFC350 FX R2 digital camera and LAS-AF software (Leica).

**Mitotic DNA replication.** To detect DNA synthesis in mitosis, cells were labelled with 10 µM EdU for 40 min followed by fixation with 4% paraformaldehyde for 20 min at room temperature and permeabilisation with PBS containing 0.5% Triton X-100 for 30 min at room temperature. Incorporated EdU was detected using the Click-iT EdU Alexa Fluor 488 Imaging Kit (Invitrogen) according to manufacturer's instructions. Cells were washed in PBS and incubated with primary antibodies for 2 h at room temperature. Then, they were washed again in PBS containing 0.1% Triton X-100 and incubated with fluorochrome-conjugated secondary antibodies (goat anti-rabbit Alexa Fluor 555 and goat anti-mouse Alexa Fluor 633, 1:1,000, Molecular Probes) for 1 h at room temperature. Coverslips were washed in PBS containing 0.1% Triton X-100 and allowed to air dry. Dried coverslips were mounted on microscope slides using the ProLong Diamond Antifade Mountant containing DAPI (ThermoFisher Scientific) and mitotic cells visualized using a wide field inverted Zeiss Axio Z1 microscope equipped with a × 63/1.4 numerical aperture objective. Images were acquired using an AxioCam MRm CCD camera (Zeiss).

**Anaphase cells analysis.** Mitotic cells were enriched by mitotic shake off and centrifugation of cells on a poly-L-lysine-coated microscope slide at 700g for 2 min. Slides were then blocked for 30 min in pre-chilled 100% methanol at −20 °C for 30 min. Slides were blocked in 10% FBS in PBS for 30 min followed by incubation with primary antibody against BLM (1:100 prepared in 0.1% FBS, C-18, Santa Cruz Biotechnology) for 1 h at room temperature. Slides were washed three times in PBS and incubated with secondary antibody for 45 min at room temperature. Slides were washed five times in PBS and mounted using Vectashield reagent containing DAPI (Vector Laboratories). Images were acquired using a DeltaVision DV Elite microscope using a 40x objective. Image analysis was carried out with FIJI (ImageJ) and Huygens Professional (Scientific Volume Imaging) software.

**DNA fibre assay.** Newly replicated DNA in H1299 or U2OS cells was labelled by addition of 25 µM CldU to the media, followed by 30 min incubation at 37 °C. Next, cells were washed three times with warm PBS and fresh media containing

250 µM IdU was added to each well. After incubation for 30 min at 37 °C, cells were harvested by trypsinization and $5 \times 10^5$ cells were re-suspended in cold PBS. Next, 7 µl of lysis buffer (200 mM Tris-HCl pH 7.4, 50 mM EDTA, 0.5% SDS) were mixed with 2 µl of cell suspension on a microscopy slide and incubated horizontally for 7 min at room temperature. The DNA was spread by tilting the slide manually at an angle of 30–45°. The air-dried DNA was fixed in methanol/acetic acid (3:1) for 10 min. Slides were rehydrated in PBS twice for 3 min and the DNA was denatured in 2.5 M HCl for 1 h at room temperature. The slides were washed several times in PBS until a pH of 7–7.5 was reached, followed by incubation in blocking solution (2% BSA, 0.1% Tween 20, PBS; 0.22 µm filtered) for 40 min at room temperature and in primary antibodies (rat anti-CldU (Abcam) and mouse anti-IdU, 1:100 (Becton Dickinson)) for 2.5 h at room temperature. After five washes in PBS-Tween (0.2% Tween 20 in PBS) for 3 min and one short wash in blocking solution, the slides were incubated with the secondary antibodies (anti-rat Alexa Fluor 555 and anti-mouse Alexa Fluor 488, 1:300, Molecular Probes) for 1 h at room temperature. Subsequently, they were washed as before, air-dried and mounted in Antifade Gold. Images were acquired as described for immunofluorescence and analysed using ImageJ software (National Healthcare Institute, USA). Fork velocity was calculated using a conversion factor of 1 µm = 2.59 kb min$^{-1}$ (ref. 36).

**Immunoblotting and fractionation.** Cells were collected by trypsinization, washed with cold PBS, re-suspended in SDS–polyacrylamide gel electrophoresis loading buffer (0.16 M Tris–HCl pH 6.8, 4% SDS, 20% glycerol, 0.01% bromophenol blue, 100 mM DTT), sonicated and boiled at 70 °C for 10 min to prepare whole-cell extracts. Fractionation of human cells was performed by incubating the pellet in buffer A (10 mM HEPES pH 7.9, 10 mM KCl, 1.5 mM MgCl$_2$, 0.34 M sucrose, 10% glycerol, 1 mM DTT, 0.05% Triton X-100, protease inhibitor cocktail) for 5 min at 4 °C to obtain the cytosol fraction. Pellet was washed and then incubated in buffer B (3 mM EDTA, 0.2 mM EGTA, 1 mM DTT, protease inhibitor cocktail) for 30 min at 4 °C to obtain nuclear fraction. Remaining pellet (chromatin fraction) was washed and re-suspended in SDS–polyacrylamide gel electrophoresis loading buffer. Equal amounts of protein (50–100 µg) were analysed by gel electrophoresis followed by western blotting. NuPAGE-Novex 10% Bis-Tris and NuPAGE-Novex 3–8% Tris-Acetate gels (Life Technologies) were run according to manufacturer's instructions. Original uncropped images of western blots used in this study can be found in Supplementary Fig. 8.

**Antibodies.** The following antibodies were used for immunoblotting: mouse monoclonal antibodies raised against BRCA1 (1:2,000, OP92, Calbiochem), BRCA2 (1:2,000, OP95, Calbiochem), GAPDH (1:30,000, 6C5, Novus Biologicals), histone H3 (1:1,000, 865R2, Invitrogen), MUS81 (1:1,000, MTA30 2G10/3, Abcam), POLD3 (1:500, 3E2, Abnova), RAD51C (1:1,000, 2H11/6, Novus Biologicals), SLX4 (a gift from Professor John Rouse, University of Dundee) and α-tubulin (1:15,000, TAT1, Cancer Research UK Monoclonal Antibody Service). Rabbit polyclonal antibodies raised against phosphorylated histone H3 Ser10 (1:1,000, Abcam), RAD51 (1:2,000, H92, Santa Cruz) and SMC1 (1:4,000, BL308, Bethyl Laboratories). Additional antibodies used for immunofluorescence detection were: mouse monoclonal antibodies raised against α-tubulin (1:2,000, TAT1, Cancer Research UK Monoclonal Antibody Service), γ-tubulin (1:2,000, GTU-88, Abcam), cyclin A (1:1,000, B-8, Santa Cruz Biotechnology), rabbit polyclonal antibodies raised against 53BP1 (1:5,000, NB100-304, Novus Biologicals), FANCD2 (1:1,000, ab2187, Abcam) and goat polyclonal antibody raised against BLM (1:100, C-18, Santa Cruz Biotechnology).

**Data availability.** The authors declare that all relevant data supporting the findings of this study are available within the article and its Supplementary Information files, or from the corresponding author upon reasonable request.

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

## Acknowledgements

We thank Dr John Rouse (University of Dundee), Dr Sheroy Minocherhomji (University of Copenhagen) and Dr Anderson Ryan (University of Oxford) for valuable reagents and/or technical suggestions. We are grateful to Dr Graham Brown and Mick Woodcock for their assistance with microscopy and FACS analyses, respectively. Work in J.S.H. laboratory is supported by funding from INCa-PLBIO 2016, ANR PRC 2016, Labex Toucan and La Ligue contre le Cancer (Equipe labellisée). Research in W.N. laboratory is supported by funding from the Medical Research Council (MRC). Research in M.T. laboratory is supported by Cancer Research UK, Medical Research Council, University of Oxford and EMBO Young Investigator Program.

## Author contributions

M.T. and X.L. designed the study and the experiments. M.T. and X.L. wrote the manuscript. X.L. performed the majority of the experiments with the help of J.Z. and S.B. R.B. carried out DAPI bridges analysis and helped with quantifications. V.B. measured DNA synthesis in mitotic cells. All authors commented on the manuscript.

## Additional information

**Competing interests:** The authors declare no competing financial interests.

DOI: 10.1038/ncomms16171   **OPEN**

# Corrigendum: MUS81 nuclease activity is essential for replication stress tolerance and chromosome segregation in BRCA2-deficient cells

Xianning Lai, Ronan Broderick, Valérie Bergoglio, Jutta Zimmer, Sophie Badie, Wojciech Niedzwiedz, Jean-Sébastien Hoffmann & Madalena Tarsounas

*Nature Communications* 8:15983 doi: 10.1038/ncomms15983 (2017); Published 17 Jul 2017; Updated 26 Oct 2017.

In this Article, there are errors in the labelling of the *y* axis in Fig. 1a and Supplementary Fig. 1b. The labels '20', '40', '60', '80' and '100' should have been '0.5', '1.0' and '1.5' in Fig. 1a and the labels '50', '100', '150' and '200' should have been '1', '2' and '3' in Supplementary Fig. 1b. The correct versions of these figures appear below as Figs 1 and 2, respectively.

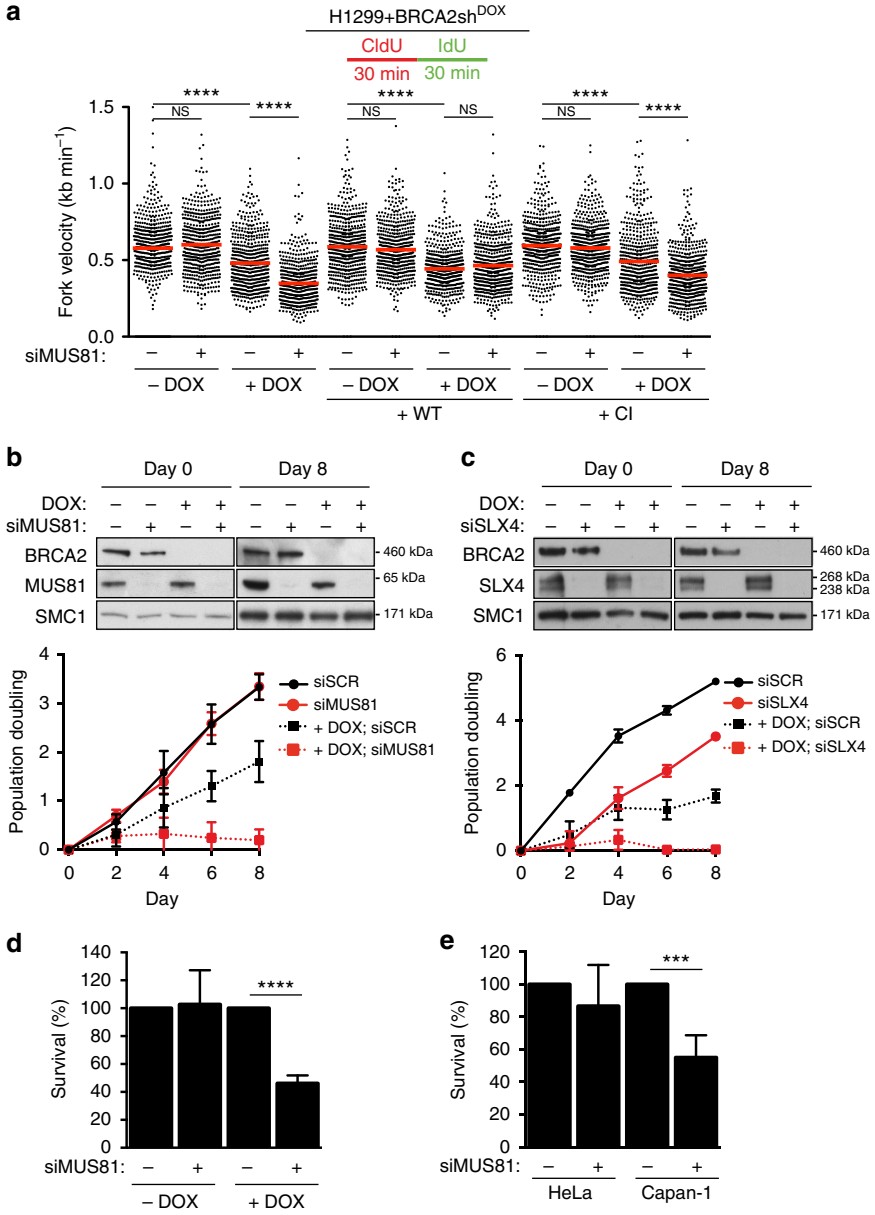

**Figure 1**

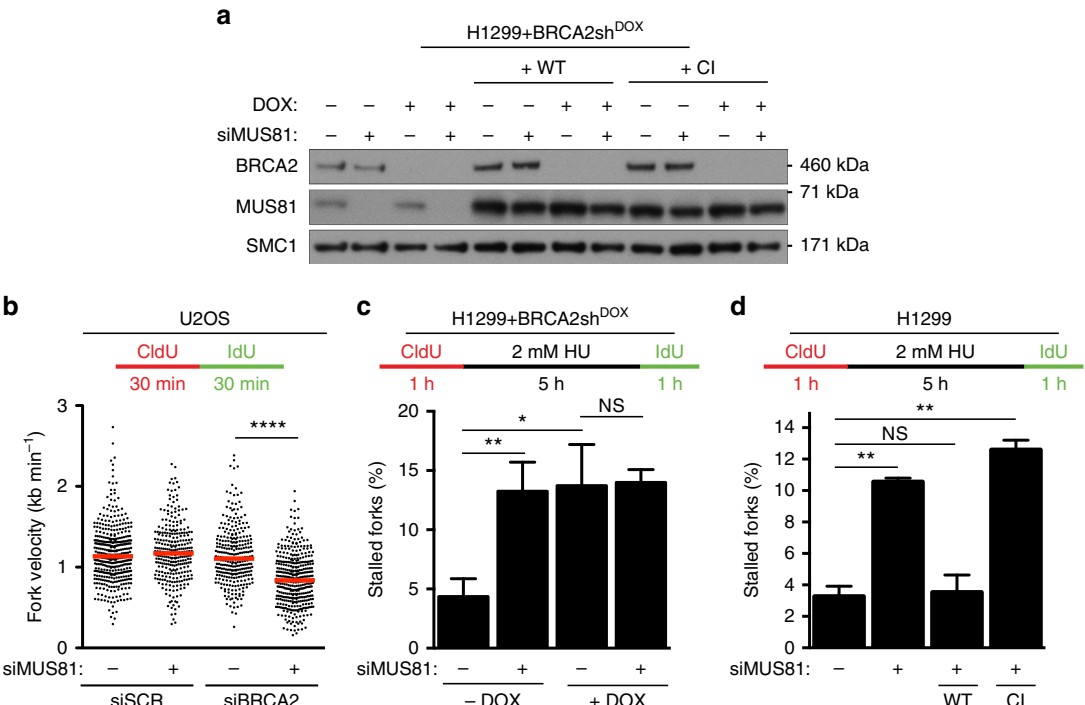

**Figure 2**

