## [Peer Review File · Nature Communications]

Reviewers' comments:

Reviewer #1 (Remarks to the Author):

Homologous recombination repair (HRR) of cytotoxic double strand breaks (DSBs) is essential to prevent genomic instability and the development of cancer. The best example is that defects in the HRR pathway, such as BRCA1/2, are associated with human cancers. BRCA2 handles the replication stress by stabilising stalled replication forks permitting cell cycle progression. MUS81 is a major nuclease, which cleaves the DNA structures in the replication forks to promote HRR in the repair of stalled replication forks. In this manuscript, Lai et al. studied whether MUS81 and BRCA2 interact in handling replication stress in order to understand the underlying mechanisms operating the pathways. The authors found that indeed MUS81 and BRCA2 function synergistically in replication fork progression and in preventing chromosome instability (or catastrophe). A double-deletion of MUS81 and BRCA2 sensitize cells to death, which may present a new approach to target MUS81 as a pharmaceutical target for treating BRCA-deficient cancers. This study shows the interaction of both the MUS81 and BRCA2 pathway in the maintenance of proper replication, mitosis and genomic stability. Overall, their findings are novel and useful for the field of DNA repair and also potentially for cancer treatment. However, several points should be addressed, which I believe can make a strong case for publication in Nat Comms.

Major comments

1. Fig 1e, f. How were the 100% determined? The P value should be provided for a comparison between -DOX/siMUS and +DOX/siMUS.
2. Fig 2. Either siPOLD3 or siMUS81 repressed the premature progression of the under-replicated DNA of BRCA2-deficient cells. However neither alone could reduce it to the control level. This suggests another mechanism to be involved. At least at this stage, a MUS81-POLD3 double deletion in BRCA2-deficient cells should be analysed.
3. Fig 3. There are several problems. (1) The authors claimed that WT-MUS81 but not CI-MUS could repress the ABs. However, this claim can be only valid if there is a similar ectopic expression level of either protein. The Western blot in the panel c does not show this convincingly. (2) It seems that the overexpression of CI-MUS81 induces more anaphase bridges (ABs) than siMUS (+CI/-DOX/+siMUS vs -DOX/+siMUS). Is the difference significant? What does this mean? (3) The P value should be given between WT/-DOX/-siMUS and IC/-DOX/-siMUS. This seems to indicate a competition between endogenous MUS and CI-MUS for repressing ABs. (4) CI/-DOX/+siMUS and +DOX/+siMUS seem to give rise to a similar ABs phenotype, suggesting that CI-MUS can override both BRCA2/MUS functions together. What does this mean? (5) The P values should be provided for -DOX/+siMUS vs +DOX/+siMUS and also for the same samples in the background of the CI overexpression.
4. The control of the overexpression level of CI-MUS and WT-MUS in Fig 4 and Fig 5 should be provided.
5. Fig 5c. It is unclear whether "G1" contains the 4N or the >4N cell population (multinucleated cells).
6. Since the authors want to make a strong point that targeting the MUS81 enzyme activity can be considered as another strategy for killing BRCA1/2 deficient cancers, they should test the feasibility at least at the cellular level.

Reviewer #2 (Remarks to the Author):

In this paper the authors demonstrate that the replication defects due to the absence of BRCA2 are dependent on MUS81 endonuclease. They show that BRCA2 and MUS81 are synthetic lethal. Consistent with these replication defects, amount of unreplicated DNA after mitosis, anaphase bridges and cytokinesis defects are increased. In addition 53BP1 foci occurring in G1 likely due to problems during the previous S phase are also increased. Most of the results are clear and well presented, even though there are problems with statistical analysis (I will explain letter). However

some data in supplemental part do not support fully the conclusions of the main figures (SupFig1, SupFig6).

One of my concern in the fact that they attribute the need for MUS81 to replication stress caused by the absence of BRCA2. However I am not sure that BRCA2 depletion by itself induces so much problems during S phase, indeed the effect of BRCA2 depletion on replication forks progression is not striking (Fig1A; ~ 28 vs ~ 20) and is not reproduced in U2OS cells (SupFig1). In addition a recent paper from Nussenzweig's lab (Chaudhuri et al Nature 2016, SupFig2) failed to see any effect on forks progression in absence of BRCA2. The synthetic lethal interaction between BRCA2 and MUS81 is new and potentially very interesting. However it is really surprising that BRCA1 and RAD51 are not SL with MUS81 (SupFig3) since anaphase bridges are also formed in their absence (Laulier et al. NAR 2011). They claim that this is due to a HR-independent role of BRCA2 but I am not sure they can conclude this since BRCA1 and RAD51 are also required for replication forks protection (Schlachter et al. Cancer Cell 2012; Hashimoto et al. NSMB 2010) like BRCA2 (Schlachter et al. Cell 2011). The authors should discuss all of this more carefully. Finally to fully establish the SL between MUS81 and BRCA2 it would have been interesting to check if cell lines with BRCA2 mutations found in patients are indeed SL with MUS81.

Another concern is that most of the results are somehow expected. It is known that depletion of BRCA2 (and also BRCA1 and RAD51) leads to problems in anaphase such as bridges (Laulier et al. NAR 2011). It is also established that replication stress leads to problems during anaphase that are dependent on MUS81 (Minocherhomji et al. Nature 2015). Therefore it is not that surprising that BRCA2 depletion leads to MUS81-dependent anaphases bridges. In addition to the issues raised above, there are more precise problems I will describe below. In conclusion I do not think that this paper is sufficiently advanced and novel for the readers of Nature Communications.

Specific comments

On figure 1A and Sup Fig 1A the number of repetitions is not clear, they say in the legend that the red line represents the average of 4 experiments meaning they pooled all the tracts lengths I guess. It would have been better to show the results of one single experiment (with the number of fibers measured in the experiment). They could show the result of one experiments and the means for all the repetitions in supplementary data. In addition I am not sure measuring CldU + IdU is the best. The length of IdU only in CldU + IdU tracts is more informative since the CldU length could be biased by activation of an origin.

Figure 1B. I am not sure they can use unpaired two-tailed t test with 3 repetitions since one need to prove that the samples need to follow a normal law

Figure 1C 1D: Would it be possible to put the average of the three independent experiments? The SD represents only the experimental variation which is not very informative.

Figure 1E: they should not normalize to 100% the BRCA2 depletion alone it is misleading.

Figure 2B. The impact on EdU incorporation when MUS81 is depleted in absence of BRCA2 is not striking (not sure this is really significant). They should change the conclusions in text.

Figure 3: problems with legends order

Figure 4: These results are clear but not the ones obtained in U2OS (SupFig6). They should comment on this.

Figure 5C: The number of cells in S phase is very low in absence of BRCA2, could it be a problem for the conclusions of the paper?

Reviewer #3 (Remarks to the Author):

In this manuscript, Lai et al. investigate the role of the MUS81 endonuclease during replication fork and mitotic repair in BRCA2-deficient cells. The authors first demonstrate that MUS81 promotes replication in the absence of BRCA2, independent of its role in fork restart. As a result, the MUS81-SLX4 complex facilitates continued proliferation of BRCA2-deficient cells. The authors next focus on the role of MUS81 in promoting mitotic repair. They show that MUS81 is important for mitotic DNA synthesis and resolution of chromatin bridges during a shortened mitosis in BRCA2-deficient cells. Furthermore, MUS81 depletion leads to an increase in anaphase bridges, multinucleation, supernumerary centrosomes, and 53BP1 foci in G1 daughter cells. All of these alterations are rescued by expression of WT MUS81 but not a catalytically inactive version.

While much of the data presented on MUS81 function at the replication fork and in mitosis is predicted based on previous work, this manuscript does address a direct relationship between MUS81 and BRCA2 and has some implications to new treatment for these tumors. More insight into the role of MUS81 in maintaining fork velocity and the synthetic lethality with BRCA2 would increase the novelty and strength of this manuscript.

Comments:

- 1) The authors speculate that MUS81 may cleave secondary structures ahead of the fork, explaining its role independent of fork restart. The authors could begin to test this by rescuing fork velocity with their MUS81 constructs.
- 2) One would expect that the survival of BRCA2 cells depends on the catalytic activity of MUS81. However, this is not directly tested. Further domain mapping using MUS81 constructs (in addition to the catalytically inactive version) capable of rescuing survival could yield additional mechanistic insights.
- 3) In fig 1d, siSLX4 appears to decrease BRCA2 levels. Can the authors comment on this?
- 4) The authors note synthetic lethality of BRCA2 and MUS81 in H1299 and U2OS cells. An expanded panel of cell lines, including BRCA2 knockout cells should be used to strengthen this result.
- 5) In fig 1f, the increase in annexin-V staining appears to be minor. Do the authors think other forms of cell death aside from apoptosis may contribute to the decrease in viability? Cell cycle profiles of these same populations at 0 and 8 days would be helpful to see when and where the cells are arresting, etc.
- 6) Does BRCA2-deficiency increase MUS81 chromatin localization in interphase or mitotic cells?

We, the authors, are grateful to the Referees for their constructive comments on our manuscript. We have addressed all the points raised by the Referees, which significantly improved our manuscript.

Point-by-point response:

Referee #1:

Homologous recombination repair (HRR) of cytotoxic double strand breaks (DSBs) is essential to prevent genomic instability and the development of cancer. The best example is that defects in the HRR pathway, such as BRCA1/2, are associated with human cancers. BRCA2 handles the replication stress by stabilising stalled replication forks permitting cell cycle progression. MUS81 is a major nuclease, which cleaves the DNA structures in the replication forks to promote HRR in the repair of stalled replication forks. In this manuscript, Lai et al. studied whether MUS81 and BRCA2 interact in handling replication stress in order to understand the underlying mechanisms operating the pathways. The authors found that indeed MUS81 and BRCA2 function synergistically in replication fork progression and in preventing chromosome instability (or catastrophe). A double-deletion of MUS81 and BRCA2 sensitize cells to death, which may present a new approach to target MUS81 as a pharmaceutical target for treating BRCA-deficient cancers. This study shows the interaction of both the MUS81 and BRCA2 pathway in the maintenance of proper replication, mitosis and genomic stability. Overall, their findings are novel and useful for the field of DNA repair and also potentially for cancer treatment. However, several points should be addressed, which I believe can make a strong case for publication in Nat Comms.

Major comments:

1. Fig 1e, f. How were the 100% determined? The P value should be provided for a comparison between -DOX/siMUS and +DOX/siMUS.

Response: We determined 100% in Fig. 1d (previously Fig. 1e) by normalizing cell survival to the corresponding scrambled siRNA (siSCR) controls, i.e. the value for '-DOX/siMUS81' was reported to '-DOX/siSCR' (set at 100% for -DOX) and the value for '+DOX/siMUS81' was reported to '+DOX/siSCR' (set at 100% for +DOX). To enable comparison between '-DOX/siMUS81' and '+DOX/siMUS81' we include below a representation of data in which both samples were normalized to the same control, '-DOX/siSCR'. *, $p = 0.03$ (unpaired two-tailed t test). Fig. 1e (previously Fig. 1f) does not contain 100%.

2. Fig 2. Either siPOLD3 or siMUS81 repressed the premature progression of the under-replicated DNA of BRCA2-deficient cells. However neither alone could reduce it to the control level. This suggests another mechanism to be involved. At least at this stage, a MUS81-POLD3 double deletion in BRCA2-deficient cells should be analysed.

Response: Previous studies (Minocherhomji et al., *Nature* 528, 2015) indicated that MUS81 cleavage initiates POLD3-dependent DNA synthesis in mitosis. Thus, MUS81 and POLD3 act in the same pathway of mitotic DNA synthesis. We have obtained new data (shown below and included in the new Fig. 2d,e) which are consistent with this model, i.e. concomitant depletion of MUS81 and POLD3 did not change significantly the frequency of FANCD2/EdU foci relative to MUS81 single depletion.

3. Fig 3. There are several problems. (1) The authors claimed that WT-MUS81 but not CI-MUS could repress the ABs. However, this claim can be only valid if there is a similar ectopic expression level of either protein. The Western blot in the panel c does not show this convincingly. (2) It seems that the overexpression of CI-MUS81 induces more anaphase bridges (ABs) than siMUS (+CI/-DOX/+siMUS vs -DOX/+siMUS). Is the difference significant? What does this mean? (3) The P value should be given between WT/-DOX/-siMUS and IC/-DOX/-siMUS. This seems to indicate a competition between endogenous MUS and CI-MUS for repressing ABs. (4) CI/-DOX/+siMUS and +DOX/+siMUS seem to give rise to a similar ABs phenotype, suggesting that CI-MUS can override both BRCA2/MUS functions together. What does this mean? (5) The P values should be provided for -DOX/+siMUS vs +DOX/+siMUS and also for the same samples in the background of the CI overexpression.

Response: (1) The stable cell lines used in Fig. 3b (also in new Fig. 1a and new Supplementary Fig. 1a) have been established by transfection of expression constructs, followed by neomycin selection. We have included a new analysis of the levels of ectopic expression of wild type (WT) and catalytically inactive (CI) MUS81 in new Supplementary Fig. 1a (and also below). The expression levels of the two proteins are comparable with each other, similarly to previous publications (e.g. Minocherhomji et al., *Nature* 528, 2015 - Extended Data Figure 4a).

(2) - (5) The p values requested by the reviewer are significant. We included them below, as a Figure for Reviewers, in reference to the graph shown in Fig. 3b. CI-MUS81 clearly has a dominant-negative effect, possibly by binding and preventing recruitment of other nucleases to anaphase bridges. A note explaining this effect is included on page 8 (top paragraph) of the revised manuscript.

(4) Biologically relevant comparisons can only be made within the groups (untreated, WT-MUS81 or CI-MUS81). As indicated above, the CI-MUS81 has a dominant-negative effect, which may explain the phenotype similarity between the groups identified by the Referee.

4. The control of the overexpression level of CI-MUS and WT-MUS in Fig 4 and Fig 5 should be provided.

Response: As above, these stable cells lines constitutively express CI and WT-MUS81. The pattern shown in new Supplementary Fig. 1a is illustrative for the levels of expression in these experiments.

5. Fig 5c. It is unclear whether "G1" contains the 4N or the >4N cell population (multinucleated cells).

Response: The G1 population shown in Fig. 5c contains 2N cells.

6. Since the authors want to make a strong point that targeting the MUS81 enzyme activity can be considered as another strategy for killing BRCA1/2 deficient cancers, they should test the feasibility at least at the cellular level.

Response: Our new data demonstrate that MUS81 siRNA-mediated inhibition in human Capan-1 cells, established from a BRCA2-deficient pancreatic tumor, showed a significant decrease in survival rates using clonogenic assays. The data shown below have now been included in the new Supplementary Fig. 3f,g.

Referee #2:

In this paper the authors demonstrate that the replication defects due to the absence of BRCA2 are dependent on MUS81 endonuclease. They show that BRCA2 and MUS81 are synthetic lethal. Consistent with these replication defects, amount of unreplicated DNA after mitosis, anaphase bridges and cytokinesis defects are increased. In addition 53BP1 foci occurring in G1 likely due to problems during the previous S phase are also increased. Most of the results are clear and well presented, even though there are problems with statistical analysis (I will explain letter). However some data in supplemental part do not support fully the conclusions of the main figures (SupFig1, SupFig6).

One of my concern in the fact that they attribute the need for MUS81 to replication stress caused by the absence of BRCA2. However, I am not sure that BRCA2 depletion by itself induces so much problems during S phase, indeed the effect of BRCA2 depletion on replication forks progression is not striking (Fig1A; ~28 vs ~20) and is not reproduced in U2OS cells (SupFig1). In addition a recent paper from Nussenzweig's lab (Chaudhuri et al Nature 2016, SupFig2) failed to see any effect on forks progression in absence of BRCA2.

Response: BRCA2 inactivation affects replication fork progression to a different extent in different cell lines. For example, as the reviewer indicates, Chaudhuri et al., *Nature* 535, 2016 find only a modest reduction in the CldU or IdU track length in *Brca2*^{-/-} vs *Brca2*^{+/+} mouse B lymphocytes (Extended data Figure 2f). In contrast, a significant reduction in replication rate relative to wild type is reported in other BRCA2-deficient human H1299 and 293T cells by Michl et al., *Nat. Struct. Mol. Biol.* 23, 2016 (Fig. 1d, e, f) or in hamster V-C8 cells by Wilhelm et al., *PNAS* 111, 2014 (Fig. 1B). In the new version of our manuscript we have strengthened the data in Fig. 1a, by examining the effect of CI MUS81 on fork velocity, as requested by Referee #3, point 1 (see below).

The synthetic lethal interaction between BRCA2 and MUS81 is new and potentially very interesting. However it is really surprising that BRCA1 and RAD51 are not SL with MUS81 (SupFig3) since anaphase bridges are also formed in their absence (Laulier et al. NAR 2011). They claim that this is a due to a HR-independent role of BRCA2 but I am not sure they can conclude this since BRCA1 and RAD51 are also required for replication forks protection (Schlacher et al. Cancer Cell 2012; Hashimoto et al. NSMB 2010) like BRCA2 (Schlacher et al. Cell 2011). The authors should discuss all of this more carefully.

Response: We propose that the reliance of BRCA2-deficient cells on MUS81 is unique among other homologous recombination deficiencies because, in addition to under-replicated DNA being carried into mitosis, BRCA2-deficient cells also lack a functional spindle assembly checkpoint. Thus, mitotic defects are detected in the absence of BRCA2 and the MUS81 requirement for survival is most stringent in these cells. We have clarified this unique aspect of BRCA2 biology on p. 6, bottom paragraph.

Finally to fully establish the SL between MUS81 and BRCA2 it would have been interesting to check if cell lines with BRCA2 mutations found in patients are indeed SL with MUS81.

Response: Data showing the effect of MUS81 siRNA-mediated inhibition in human Capan-1 cells are included in our response to Referee #1, point 6.

Another concern is that most of the results are somehow expected. It is known that depletion of BRCA2 (and also BRCA1 and RAD51) leads to problems in anaphase such as bridges (Laulier et al. NAR 2011). It is also established that replication stress leads to problems during anaphase that are dependent on MUS81 (Minocherhomji et al. Nature 2015). Therefore it is not that surprising that BRCA2 depletion leads to MUS81-dependent anaphases bridges. In addition to the issues raised above, there are more precise problems I will describe below. In conclusion I do not think that this paper is sufficiently advanced and novel for the readers of Nature Communications.

Response: This view is not shared by the other Referees, who consider that the synergy between MUS81 and BRCA2 in genome integrity and mitotic chromosome segregation, leading to a synthetic lethal interaction between the two genes, is novel and likely to have a significant impact on the DNA repair field and cancer therapeutics. We have added the reference to Laulier et al., NAR 2011 to the revised version of our manuscript (page 8, top). We agree that this reference is

relevant to our paper, however our work is distinct in that we identify the mechanism for anaphase bridges resolution, specific to BRCA2-depleted cells.

Specific comments:

On figure 1A and Sup Fig 1A the number of repetitions is not clear, they say in the legend that the red line represents the average of 4 experiments meaning they pooled all the tracts lengths I guess. It would have been better to show the results of one single experiment (with the number of fibers measured in the experiment). They could show the result of one experiments and the means for all the repetitions in supplementary data.

Response: In the new Supplementary Tables 1 and 2 of the revised manuscript, we show both the number of fibers and the mean fiber track length measured in each experiment ($n=3$ for Fig. 1a and $n=3$ for Supplementary Fig. 1b).

			Total number of fibers quantified	Mean fiber track length (μm)		
				Experiment		
				1	2	3
	- DOX	siSCR	524	12.72	14.56	15.72
		siMUS81	558	12.56	13.32	16.63
	+ DOX	siSCR	580	8.42	12.14	13.50
		siMUS81	542	6.72	9.76	8.31
+ WT	- DOX	siSCR	587	12.34	14.55	14.14
		siMUS81	543	11.99	13.45	14.71
	+ DOX	siSCR	550	9.15	9.94	12.44
		siMUS81	589	9.80	10.02	12.73
+ CI	- DOX	siSCR	544	13.15	14.37	14.22
		siMUS81	565	12.64	13.74	14.26
	+ DOX	siSCR	552	10.95	12.45	10.82
		siMUS81	600	8.50	10.39	9.05

Supplementary Table 1. Total number of fibers and mean fiber track length (μm) quantified in Fig. 1a.

		Total number of fibers quantified	Mean fiber track length (μm)		
			Experiment		
			1	2	3
- siBRCA2	siSCR	454	23.22	25.81	27.30

	siMUS81	309	25.82	27.96	26.16
+ siBRCA2	siSCR	319	27.51	27.13	22.74
	siMUS81	421	18.14	19.36	18.65

Supplementary Table 2. Total number of fibers and mean fiber track length (μm) quantified in Supplementary Fig. 1b.

In addition I am not sure measuring CldU + IdU is the best. The length of IdU only in CldU + IdU tracts is more informative since the CldU length could be biased by activation of an origin.

Response: Measuring fork velocity as CldU+IdU is routinely used in publications, as an example please see Michl et al., *Nat. Struct. Mol. Biol.* 23, 2016.

Figure 1B. I am not sure they can use unpaired two-tailed t test with 3 repetitions since one need to prove that the samples need to follow a normal law

Response: As above, the unpaired two-tailed *t*-test is routinely used in publications, as an example please see Michl et al., *Nat. Struct. Mol. Biol.* 23, 2016.

Figure 1C 1D: Would it be possible to put the average of the three independent experiments? The SD represents only the experimental variation which is not very informative.

Response: Population doubling assays carry an inherent level of variability and therefore results from one experiment (with technical triplicates) are frequently shown in publications. As examples, please see Evers et al., *Clin. Cancer Res.* 2008 and 2010; Boersma et al., *Nature* 521, 2015 (Fig. 1c and Extended data Fig. 5h) or Zimmer et al., *Mol. Cell* 6, 2016. The results presented in our manuscript have been reproduced at least in two independent experiments (each in technical triplicates) with similar outcomes.

Figure 1E: they should not normalize to 100% the BRCA2 depletion alone it is misleading.

Response: Please see our response to Referee #1, point 1. This mode of representation is commonly used in publications. As examples, please see Kais et al., *Cell Rep.* 15, 2016 (Fig. 2A) and Mateos-Gomez et al., *Nature* 218, 2015.

Figure 2B. The impact on EdU incorporation when MUS81 is depleted in absence of BRCA2 is not striking (not sure this is really significant). They should change the conclusions in text.

Response: The *p* value calculated using an unpaired two-tailed *t*-test is 0.0184, therefore significant.

Figure 3: problems with legends order

Response: We apologize for the ordering error in Fig. 3 legend, which has now been rectified.

Figure 4: These results are clear but not the ones obtained in U2OS (SupFig6). They should comment on this.

Response: We performed additional experiments which demonstrate statistically significant effects of MUS81 depletion in U2OS cells. The data below, which are also included in the new Supplementary Fig. 7d of the revised manuscript, show that MUS81 siRNA causes supernumerary centrosomes in BRCA2-deficient U2OS human cells, similarly to H1299 cells shown in Fig. 4d.

Figure 5C: The number of cells in S phase is very low in absence of BRCA2, could it be a problem for the conclusions of the paper?

Response: The Referee is correct in pointing out that the frequency of S-phase cells is low in BRCA2-deficient cells. We observe even lower S-phase frequencies when MUS81 is depleted in these cells. One possible explanation for this effect is that some of these cells escape G1 arrest and enter S-phase where they die and are eliminated, consistent with the observed synthetic lethal interaction between *BRCA2* and *MUS81*. Since the S-phase cell reduction (Fig. 5c) is detected in our assays concomitantly with cell cycle arrest and apoptosis, we consider that it is not in conflict with the conclusions of our paper.

Referee #3:

In this manuscript, Lai et al. investigate the role of the MUS81 endonuclease during replication fork and mitotic repair in BRCA2-deficient cells. The authors first demonstrate that MUS81 promotes replication in the absence of BRCA2, independent of its role in fork restart. As a result, the MUS81-SLX4 complex facilitates continued proliferation of BRCA2-deficient cells. The authors next focus on the role of MUS81 in promoting mitotic repair. They show that MUS81 is important for mitotic DNA synthesis and resolution of chromatin bridges during a shortened mitosis in BRCA2-deficient cells. Furthermore, MUS81 depletion leads to an increase in anaphase bridges, multinucleation, supernumerary centrosomes, and 53BP1 foci in G1 daughter cells. All of these alterations are rescued by expression of WT MUS81 but not a catalytically inactive version.

While much of the data presented on MUS81 function at the replication fork and in mitosis is predicted based on previous work, this manuscript does address a direct relationship between MUS81 and BRCA2 and has some implications to new treatment for these tumors. More insight into the role of MUS81 in maintaining fork velocity and the synthetic lethality with BRCA2 would increase the novelty and strength of this manuscript.

Comments:

1) *The authors speculate that MUS81 may cleave secondary structures ahead of the fork, explaining its role independent of fork restart. The authors could begin to test this by rescuing fork velocity with their MUS81 constructs.*

Response: We agree with the Referee that rescuing fork velocity represents an entry point into understanding the mechanism of MUS81 action during replication, in the absence of BRCA2. We therefore performed fiber assays using cells expressing WT- and CI-MUS81 and established that ectopically expressed WT-MUS81, but not CI-MUS81 can rescue the fork slowdown induced by MUS81 siRNA in BRCA2-deficient cells (new Fig. 1a below, $n=3$; lanes 7-8 vs 11-12).

2) One would expect that the survival of BRCA2 cells depends on the catalytic activity of MUS81. However, this is not directly tested. Further domain mapping using MUS81 constructs (in addition to the catalytically inactive version) capable of rescuing survival could yield additional mechanistic insights.

Response: Our new data (below and included in Supplementary Fig. 2a,b of the revised manuscript) demonstrate that ectopic expression of WT-MUS81, but not CI-MUS81, can complement the proliferation defect induced by MUS81 depletion in BRCA2-deficient cells. This suggests that the nuclease activity of MUS81 is required for cell survival in the absence of BRCA2.

3) In fig 1d, siSLX4 appears to decrease BRCA2 levels. Can the authors comment on this?

Response: The decrease in BRCA2 levels upon SLX4 siRNA treatment is not consistently observed. To illustrate this point, we include below immunoblots of cell extracts obtained from an experiment different from the one in previous Fig. 1d, which don't show substantial reduction in BRCA2 expression.

4) The authors note synthetic lethality of BRCA2 and MUS81 in H1299 and U2OS cells. An

expanded panel of cell lines, including BRCA2 knockout cells should be used to strengthen this result.

Response: We have demonstrated that MUS81 siRNA-mediated inhibition results in a significant reduction of BRCA2-deficient cell survival using clonogenic assays in four different human cell lines: H1299 (Fig. 1d), U2OS (Supplementary Fig. 3c), Calu-6 (below and new Supplementary Fig. 3d,e) and Capan-1 (our response to Referee #1, point 6 and Supplementary Fig. 3f,g).

5) In fig 1f, the increase in annexin-V staining appears to be minor. Do the authors think other forms of cell death aside from apoptosis may contribute to the decrease in viability? Cell cycle profiles of these same populations at 0 and 8 days would be helpful to see when and where the cells are arresting, etc.

Response: We agree with the Referee that other forms of cell death apart from apoptosis are likely involved and have included a comment on page 6, bottom paragraph. The cell cycle profiles do not change significantly with time.

6) Does BRCA2-deficiency increase MUS81 chromatin localization in interphase or mitotic cells?

Response: To evaluate MUS81 chromatin localization, we carried out fractionation experiments of asynchronous, G2 and early mitotic cells (prometaphase). G2 cells were isolated following incubation with the Cdk1 inhibitor RO-3306 (9 μM) for 16 hours and prometaphase cells were collected following release in fresh media for 20 minutes. Upon cell fractionation, we observed increased MUS81 chromatin localization in BRCA2-deficient mitotic cells (below and new Fig. 2c).

Reviewers' comments:

Reviewer #1 (Remarks to the Author):

1. My original comment #2 was "neither siPOLD3 or siMUS81 alone repressed the premature progression to the control level, suggesting another mechanism (perhaps a major one) to be involved". However, this part has not been addressed or discussed.
2. (original #5) Fig 5. It is still unclear to me whether the "G1" analysis included any multinucleated (4N or >4N) cells derived from mitotic slippage. Where are these cells (in "G1-like"?) and what is the 53BP1 status of these cells?
3. (original #6) The authors tested the knock-down of the MUS81 protein (unfortunately not the activity) to kill BRCA2-deficient human cancer cells. These data are important and should be included in the main figure set.

Reviewer #2 (Remarks to the Author):

After reading the responses of the authors to the reviewers as well as the new version of the manuscript, I can see that the authors made considerable efforts to improve their manuscript (for instance by expanding the DNA fibers analysis and adding more details on the repetitions performed in a table). The results obtained in BRCA2 deficient cells validate clearly the SL between MUS81 and BRCA2 as a proof of concept for future therapies. I also liked the proposed role of BRCA2 in spindle assembly checkpoint that could explain the specificity of SL of MUS81 with BRCA2. It provides an interesting new finding which was not present in the original version of the manuscript. Altogether they addressed well all my comments even though I still disagree with the measure of DNA fibers (but I must admit it will not change the conclusions). Concerning the statistical tests I think they need to be careful with their conclusions when differences are not huge (see Figure 1B for example). It is especially important when sample sizes are small like here, however most of the effects they observed are really clear therefore it is not a problem for the conclusions. Altogether I think that this paper now deserves to be published in Nature Communications.

Reviewer #3 (Remarks to the Author):

The authors have addressed a number of the experimental concerns by providing new data on replication for progression and lethal interactions between BRCA2 and Mus81. Still, there remains a number of weaknesses with the study. As stated by the second Reviewer and myself, minimal novelty exists in the study. The possible synthetic lethal interaction between Mus81 and BRCA2 is interesting, but not that compelling either. Only a partial loss of viability was observed in CAPAN-1 cells following siMus81. It is not clear how this would compare to PARPi or other agents that selectively kill BRCA2 mutant cells. An additional weakness is the poorly explained specificity of the Mus81 interaction with BRCA2 in comparison to Rad51 or BRCA1. This is attributed to loss of the spindle assembly checkpoint in BRCA2 mutant cells, without any proof that it is the underlying basis. BRCA1 has also been reported as necessary for the spindle assembly checkpoint (Wang et al. PNAS 2004, PMID 15563594). If loss of the SAC is involved, there are means to test this. The impact of Mus81 on fork progression in BRCA2 deficient cells is also not sufficiently explained. Collectively, this results in a descriptive paper that has some value, but does not represent a significant advance.

Referee #1:

1. My original comment #2 was “neither siPOLD3 or siMUS81 alone repressed the premature progression to the control level, suggesting another mechanism (perhaps a major one) to be involved”. However, this part has not be addressed or discussed.

Response: This comment by Referee #1 was in connection to Fig. 2 of our original submission, in which we addressed the contribution of POLD3 and MUS81 to mitotic DNA synthesis in BRCA2-deficient cells. This process is a consequence of premature progression in mitosis with under-replicated DNA. In these experiments we did not detect a reduction to control levels with either siPOLD3 or siMUS81, as indicated by the Referee. Therefore we performed the MUS81-POLD3 double depletion in BRCA2-deficient cells, as suggested by the Referee, and analyzed the frequency of FANCD2/EdU foci relative to MUS81 single depletion (data are shown in new Figure 2d,e). These experiments are detailed in the Reply to Reviewers accompanying our submission of March 29, 2017 (Referee #1, comment no. 2). These results suggested the existence, but did not identify a new mechanism of DNA synthesis initiation alternative to the MUS81-POLD3 pathway. In the revised version of our manuscript we speculate that other nucleases may promote premature progression and DNA synthesis in mitosis (p.7 bottom and p.8 top). However, their identification is beyond the scope of the current paper.

2. (original #5) Fig 5. It is still unclear to me whether the “G1” analysis included any multinucleated (4N or >4N) cells derived from mitotic slippage. Where are these cells (in “G1-like”?) and what is the 53BP1 status of these cells?

Response: Our G1 analysis presented in Fig. 5a,b did not include multinucleated cells, as the majority of these cells were cyclin A-positive (i.e. not in the G1 stage of the cell cycle). An explanatory note is included on p.9 (line 4 from top).

3. (original #6) The authors tested the knock-down of the MUS81 protein (unfortunately not the activity) to kill BRCA2-deficient human cancer cells. These data are important and should be included in the main figure set.

Response: In the revised version of our manuscript, we have included the survival data with human Capan-1 cancer cells in Fig. 1e. The apoptosis data have been moved to Supplementary Fig. 3g. The relevant text alterations are highlighted in blue.